# Multimodal and Multilingual Embeddings for Large-Scale Speech Mining

**Paul-Ambroise Duquenne**
Facebook AI Research
padqn@fb.com

**Hongyu Gong**
Facebook AI Research
hygong@fb.com

**Holger Schwenk**
Facebook AI Research
schwenk@fb.com

## Abstract

We present an approach to encode a speech signal into a fixed-size representation which minimizes the cosine loss with the existing massively multilingual LASER text embedding space. Sentences are close in this embedding space, independently of their language and modality, either text or audio. Using a similarity metric in that multimodal embedding space, we perform mining of audio in German, French, Spanish and English from Librivox against billions of sentences from Common Crawl. This yielded more than twenty thousand hours of aligned speech translations. To evaluate the automatically mined speech/text corpora, we train neural speech translation systems for several languages pairs. Adding the mined data, achieves significant improvements in the BLEU score on the CoVoST2 and the MUST-C test sets with respect to a very competitive baseline.

Our approach can also be used to directly perform speech-to-speech mining, without the need to first transcribe or translate the data. We obtain more than one thousand three hundred hours of aligned speech in French, German, Spanish and English. This speech corpus has the potential to boost research in speech-to-speech translation which suffers from scarcity of natural end-to-end training data. All the mined multimodal corpora will be made freely available.

## 1 Introduction

Neural approaches have evolved to the de-facto standard in machine translation (MT), namely text-to-text and speech-to-text translation (Berard et al., 2016; Weiss et al., 2017), and more recently speech-to-speech translation (Jia et al., 2019). While there is very promising research on unsupervised MT, e.g. (Lample et al., 2018; Artetxe et al., 2017), or initialization of parts of the MT system by approaches trained in a self-supervised manner, labeled data remains extremely useful to achieve best performance. This labeled data is either used to directly train the MT system end-to-end, if enough data is available, or to fine-tune an MT system which parts have been initialized with some self-supervised methods. This type of approaches is commonly used in speech-to-text translation, see for example (Tran et al., 2020; Li et al., 2020).

In the case of text-to-text (t2t) machine translation, the labeled data consists of source/target sentence pairs, commonly named bitexts or parallel data. There are several international organisations which produce human translation in several languages, best known are probably the European Commission and the United Nations. These human translations have been collected and are distributed to the MT community, e.g. the well known Europarl (Koehn, 2005) and UN corpora (Ziemski et al., 2016). There is also a large body of research which aims in finding existing translation in huge collections of monolingual texts, which are abundant on the the Internet for many languages. This bitext mining has provided huge amounts of parallel data, in particular the ParaCrawl project (Esplà et al., 2019) or the CCMatrix initiative (Schwenk et al., 2021). This data enabled training a multilingual t2t

MT systems for hundred languages (Fan et al., 2020). We are not aware of any attempts to mine for speech translation data.

Training speech-to-text (`s2t`) MT systems requires audio in the source language and its translation in the target language. To the best of our knowledge, this type of data is mainly generated by humans volunteers: speech is first transcribed and then translated into a another language – a time and labour extensive process. When translations are not provided at the sentence level, forced alignments are necessary. These methodologies helped creating well-known speech translation corpora, like Must-C (Di Gangi et al., 2019a), CoVoST (Wang et al., 2020a) and EuroparlST (Iranzo-Sánchez et al., 2020).

Finally, speech-to-speech translation (`s2s`) is faced with extreme scarcity of labeled data, i.e. speech in the source language and the spoken translation in the target language. Previous work on speech-to-speech translation often use synthesized speech targets to overcome this lack of labeled data. We are only aware of one `s2s` training corpus (Wang et al., 2021) which provides speech-to-speech alignments for 16 source languages with 5 target languages. However, the target aligned speech is oral interpretation of the source speech. The quality of these oral interpretations for speech-to-speech translation have not been evaluated yet.

In this work, we build on recent advances to perform bitext mining based on a similarity measure in a fixed-size sentence embedding space (Schwenk, 2018; Artetxe and Schwenk, 2019; Feng et al., 2020). The underlying idea of those approaches is to learn a multilingual sentence embedding with the property that sentences with similar meaning, in the same or a different language, are close in the embedding space. Then, mining for translations can be performed by calculating the cosine distance, or a margin-based criterion, in this multilingual embedding space for a large collection of sentences in several languages. Sentence pairs with a distance below a threshold are considered to be parallel. We extend this idea to the speech modality and present a fixed-size embedding of a speech input, of arbitrary length. Our current audio encoder supports speech in five languages, namely English, French, Spanish, German and Russian. We apply a teacher-student training approach which yields speech embeddings which are compatible with an existing multilingual text encoder, namely the freely available LASER encoder. This means that we can mine speech input against the 80 languages which are supported by LASER, as used in the bitext mining project CCMatrix.

The contributions of this paper are as follows:

- We explore several approaches to learn multimodal (speech/text) representations for several languages;
- We use these multimodal embeddings to mine more than twenty thousand hours of speech in four languages from Librivox against texts from Common Crawl;
- We add the automatically mined data to train speech-to-text translation systems and obtain significant improvements with respect to a very strong baseline;
- We also provide a proof of concept that our approach can be used to directly perform speech-to-speech mining, without the need to transcribe and translate the input speech;

The paper is structured as follows. In the next section, we first summarize related work. In Section 3 we describe our approach in detail, followed by a detailed experimental evaluation in Section 4. The paper concludes with a summary and directions of future research.

## 2 Related work

**Bitext mining.** There is a large body of research on bitext mining from monolingual resources, i.e. finding parallel texts in a source and target language. A large variety of approaches has been explored like relying on document meta-information (Resnik, 1999), cross-lingual document retrieval (Munteanu and Marcu, 2005) or machine translation and information retrieval (Abdul-Rauf and Schwenk, 2009; Bouamor and Sajjad, 2018), just to name a few. Our approach for speech mining is inspired by recent work on massively multilingual sentence embeddings, i.e. fixed-size text representations with the property that similar sentences written in different languages are close in the embedding space. Mining is then performed by simple exhaustive comparison of sentences in different languages, using either a threshold on the cosine distance (Schwenk, 2018) or a margin criterion (Artetxe and Schwenk, 2018; Yang et al., 2019a). The multilingual sentence representations can be learned with neural machine translation framework (España-Bonet et al., 2017; Schwenk and

Douze, 2017; Artetxe and Schwenk, 2019; Kvapilíková et al., 2020). Recent works focus on dual encoders, for example (Yang et al., 2019a; Reimers and Gurevych, 2019; Yang et al., 2019b; Feng et al., 2020). Finally, knowledge distillation was proposed to extend existing multilingual sentence embeddings to new languages (Reimers and Gurevych, 2020). Our approach is similar to that work: we extend an existing multilingual sentence embedding to the speech modality.

**Fixed-size speech representations**. Fixed-size speech representations have been mainly studied at the word level for different specific tasks, ranging from spoken term detection, speech pattern discovery, to speech segmentation into words. Different approaches have been studied to extract a fixed-size representation from speech input. Holzenberger et al. (2018) introduced a method to extract a fixed-size vector from speech inputs using Gaussian downsampling, without the need of any training. Holzenberger et al. (2018); Chung et al. (2016) introduced an auto-encoder approach based on recurrent neural networks to extract a fixed-size vector between the encoder and the decoder. Some other works (Settle and Livescu, 2016; Riad et al., 2018; Thiolliere et al., 2015) train Siamese networks with a contrastive loss to build a fixed-size representation. Audhkhasi et al. (2017) studies a keyword search task, building fixed-size representations for audio and words before inputting these representations to a third neural network.

At the sentence level, text-audio sentiment analysis often intrinsically introduces a fixed-size cross-modal representation before classifying the input, as in (Yang et al., 2020; Tsai et al., 2019). However, such works do not focus on speech/text alignments, but rather take advantage of information coming from both modalities. Khurana et al. (2020) focuses on speech representation learning with speech translation data and a contrastive loss at the sentence level. The model is first evaluated on a retrieval task but not used for large-scale speech translation mining, the speech encoder is rather used for a phone recognition task. Harwath et al. (2018); Merkx et al. (2019); Ilharco et al. (2019); Harwath et al. (2019); Monfort et al. (2021) build joint speech/visual embedding spaces at the sentence level and are evaluated with a retrieval task.

**Speech mining**. We are not aware of any approach that performed mining of speech against either texts or even speech in another language. Speech input could be first transcribed by an existing speech recognition system. There are several works which use these automatic transcriptions, often named *"pseudo labels"*, to improve speech recognition. However, we are not aware that automatically transcribed speech was used as input to a text-to-text mining approach with the goal to create a speech translation corpus. One of the challenges are the different conventions: transcriptions are usually all lower case, without punctuation and numbers are spelled out as multiple words. To the best of our knowledge, all existing s2t corpora are using human translations, either volunteers translating sentences, or doing force-alignment when translations are not provided at the sentence level. Must-C (Di Gangi et al., 2019a) provides more than 385 hours of English speech translated in 8 languages, CoVoST (Wang et al., 2020a) provides speech translations in 15 En-X directions and 22 X-En directions, some languages are presented in Table 1, and EuroparlST (Iranzo-Sánchez et al., 2020) provides 30 different speech translation directions from 6 European languages.

## 3   Speech encoder training

Training a multimodal audio/text fixed-size embedding could be motivated by recent research on training Siamese networks with a contrastive loss, e.g. (Feng et al., 2020). Instead of two text encoders, one would use one speech encoder and one text encoder. However, since both encoders are trained from scratch, such a procedure would probably require a large amount of labeled multimodal training data. Instead, we apply a teacher/student training framework: we use an existing text encoder as teacher and train an audio encoder to minimize the cosine loss between the two encoder outputs. This architecture is summarized in Figure 1. It can be trained with two types of labeled data:

- **Speech transcriptions**: both encoders use input in the same language, but differ in the modality;

- **Speech translations**: since the text encoder is multilingual, we can also minimize the cosine loss of the speech embedding with respect to its written translation in one of the languages supported by the text encoder.

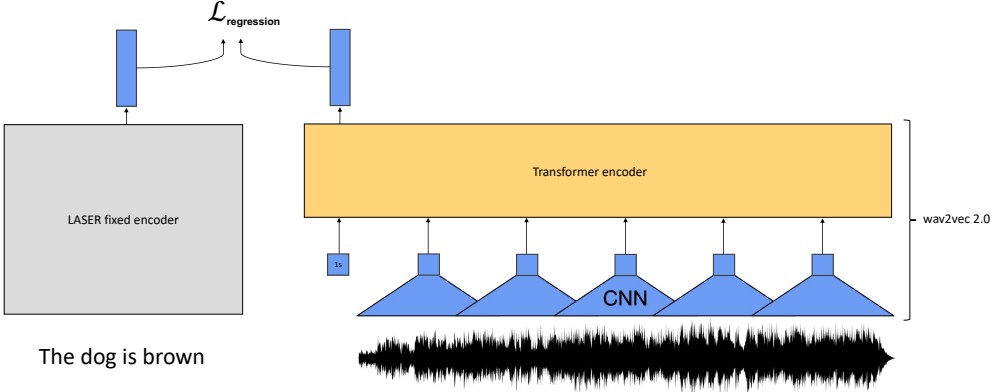

Figure 1: Architecture of our teacher/student approach to train a speech encoder based on the existing LASER sentence encoder.

Concretely, we use the multilingual LASER sentence encoder which is freely available[1] and which was successfully used in large-scale text mining approaches (Schwenk et al., 2021). A thorough comparison with other multilingual sentence encoders is left for future research, in particular LaBSE (Feng et al., 2020). The LASER encoder is fixed during teacher/student training. Our speech encoder is based on wav2vec 2.0 architecture (Baevski et al., 2020), a speech encoder which learns high-quality speech representations from unlabelled speech data. It is composed of a CNN feature encoder stacked with a transformer encoder. In recent work, wav2vec 2.0 was extended to multiple languages: the XLSR model (Conneau et al., 2020).[2] We used the XLSR pre-trained feature encoder, without updating its weights, and fine-tuned the weights of XLSR transformer encoder to obtain our fixed-size audio representation. During fine-tuning, the feature encoder representations are masked with a strategy similar to SpecAugment (Park et al., 2019) as introduced in wav2vec 2.0 paper.

We explored several methods to get a fixed-size speech representation, like max- or mean-pooling of the encoder outputs. Best performance was obtained using the output of the transformer encoder corresponding to a particular begin-of-sentence (BOS) vector. This BOS vector is simply a vector filled with ones (1.0), added at the beginning of the feature sequence in wav2vec2.0 architecture. This method is inspired by BERT (Devlin et al., 2019) sentence representations for text, that are often extracted using a CLS token at the beginning of the input sentence, e.g. (Reimers and Gurevych, 2019).

## 3.1 Encoder evaluation

In order to compare different versions of our speech encoder, an evaluation framework is needed. The ultimate goal is to mine speech translation and to show improvements when training s2t MT systems. However, this is computationally rather expensive and we only apply it for some selected encoders (see Section 4). To evaluate a standalone speech encoder, we propose to use multimodal similarity search. Given a multimodal test set $(a_i, t_i)_{1,...,N}$ with $a_i$ being the audio file and $t_i$ its corresponding text, we encode the speech and texts. We normalize the texts by removing quotes encapsulating whole sentences and lower casing them. For each speech encoding $a_i^E$, we then search the closest text embedding $t_j^E$, counting an error if it is not the expected one. Multilingual similarity search is commonly used in many papers on sentence embeddings (Artetxe and Schwenk, 2019; Hu et al., 2020; Feng et al., 2020). In particular, we adopt the margin based similarity proposed in Artetxe and Schwenk (2018) which was reported to outperform simple cosine similarity. The margin $sim(x, y)$ between two embeddings $x$ and $y$ is defined as the difference between the cosine similarity

---

[1]https://github.com/facebookresearch/LASER
[2]https://github.com/pytorch/fairseq/tree/master/examples/wav2vec

between $x$ and $y$, and the average cosine similarity of its nearest neighbors in both directions:

$$\text{sim}(x,y) = \cos(x,y) - \left( \sum_{z \in \text{NN}_k(x)} \frac{\cos(x,z)}{2k} + \sum_{z \in \text{NN}_k(y)} \frac{\cos(y,z)}{2k} \right) \quad (1)$$

where $NN_k(x)$ are the nearest neighbors of $x$. We use the Dev and Test set of the CoVoST2 corpus (Wang et al., 2020a) which statistics are summarized in Table 1.

Table 1: Statistic of CoVost2 speech translation corpus used to train and evaluate the speech encoders.

|  | En | | | De | | | Es | | | Fr | | | Ru | | |
|---|---|---|---|---|---|---|---|---|---|---|---|---|---|---|---|
|  | Train | Dev | Test | Train | Dev | Test | Train | Dev | Test | Train | Dev | Test | Train | Dev | Test |
| Audio [hours] | 430 | 26 | 25 | 184 | 21 | 22 | 113 | 22 | 23 | 264 | 22 | 23 | 18 | 10 | 11 |
| #sentences | 289k | 16k | 16k | 128k | 14k | 14k | 79k | 13k | 13k | 207k | 15k | 15k | 12k | 6k | 6k |

### 3.2 Single multilingual speech encoder

All our speech encoders are trained and evaluated on the CoVoST dataset (see Table 1) released under CC0 license. CoVoST is a large-scale multilingual speech translation corpus based on Common Voice (Ardila et al., 2019). In this work, we focused on five spoken languages: English, German, French, Spanish and Russian. For each audio input language, we explored different textual training targets, namely the transcriptions encoded with LASER and the English translation encoded with LASER. We use the German translations as a teacher for English speech data. We call them respectively, the *"transcription teacher"* and the *"translation teacher"*. We also train on both, i.e. using transcriptions and translations as teachers. Future work may use larger existing training sets for speech recognition.

In this section, we first train one multilingual speech encoder for all five languages. To handle unbalanced training data between languages, speech sentences are sampled according to a multinomial distribution with probabilities $\{q_i\}_{i=1,..,N}$:

$$q_i = \frac{p_i^\alpha}{\sum_{j=1}^{N} p_j^\alpha} \quad \text{with} \quad p_j = \frac{n_j}{\sum_{k=1}^{N} n_k} \quad (2)$$

In the following experiments, we use $\alpha = 0.2$. For all training methods, we take the checkpoint with the lowest validation loss. The learning rate to finetune XLSR transformer is set to $10^{-4}$, and training was performed on 24 Tesla V100 GPUs.

Our multimodal similarity search results are summarized in Table 2. In the top block of results, we first report the multimodal similarity error rates between the speech embeddings and the text

Table 2: Similarity search results (error rates) for the different training methods with a multilingual speech encoder

| Teacher mode | en | de | fr | es | ru |
|---|---|---|---|---|---|
| **A) Search audio against transcriptions** | | | | | |
| A.1    Transcriptions | 2.70 | 1.03 | 0.79 | 0.57 | 25.63 |
| A.2    Translations | 3.25 | 1.93 | 1.40 | 0.89 | 28.32 |
| A.3    Both | 3.01 | 1.21 | 0.91 | 0.64 | 36.19 |
| **B) Search audio against translations (en)** | | | | | |
| B.1    Transcriptions | - | 3.58 | 2.31 | 1.79 | 30.46 |
| B.2    Translations | - | 4.06 | 2.57 | 1.88 | 31.65 |
| B.3    Both | - | 3.36 | 2.05 | 1.66 | 40.54 |
| **C) Search transcriptions against translations (en)** | | | | | |
| n/a | - | 1.96 | 0.97 | 1.00 | 1.05 |

embeddings of the human transcriptions, for the three training methods. The error is below 1% for German, French and Spanish when the training and evaluation criterion are the same (row A.1). The performance on Russian is significantly worse: about 25% error rate. Not surprisingly, the error rates are higher when using the embeddings of translations into English as targets (row A.2), but also when using both (row A.3).

We then switch to similarity search of speech against the translation into English (block B). These results are relevant to our use case of speech mining (see Section 4). Overall, the error rates are about twice as high. Surprisingly, performance is slightly better when using transcriptions as the teacher (row B.1) than translations (row B.2), although this corresponds to the evaluation criteria. Best results are obtained when using both (row B.3). Finally, as lower bound, we calculate the similarity error between the speech transcriptions and their translations (block C). Both source and target are sentences, i.e. no audio encoder is used. The error rates is about 1% for French, Spanish and Russian, and 2% for German. Compared to these numbers, the performance of our multilingual speech encoder (row B.3) seems to be very good (with exception of Russian).

### 3.3 Separate speech encoders per language

We now switch to separate speech encoders, trained with both translations and transcriptions as teachers. As can be seen in Table 3, we observe a huge improvement for the Russian speech encoder: the error rate against the English translations went down from more than 30% to 6.9%.

Table 3: Similarity search results for separate speech encoders (transcription+translation teachers)

| | en | de | fr | es | ru |
|---|---|---|---|---|---|
| Search audio against transcriptions | 2.72 | 1.44 | 0.84 | 0.57 | 3.46 |
| Search audio against translations (en) | - | 3.73 | 1.86 | 1.78 | 6.86 |
| Search transcriptions against translations (en) | - | 1.96 | 0.97 | 1.00 | 1.05 |

The error rate for French decreased from 2.05% to 1.86%, but those for Spanish and German are slightly higher than with the multilingual speech encoder. Our experimental evidence seems to indicate that low-resource languages, e.g. Russian in our study, are better handled by an individual speech encoder.

We also studied the quality of the speech encoder in function of the training data (transcriptions only, Fig 2). As expected, more data gives better performance, but the curves flattens out quickly and good performance is already achieved for relatively small amounts of training data.

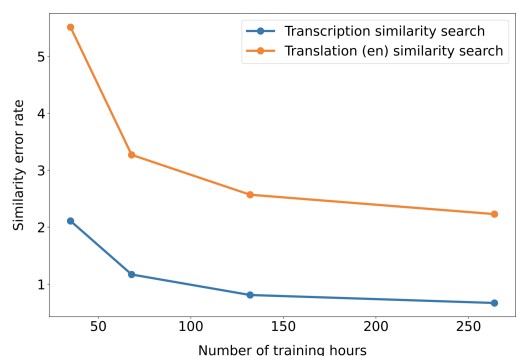

Figure 2: Similarity error rate on French speech for different training sizes (transcription teacher).

## 4 Speech mining and translation

We now apply the encoder to mine unlabeled raw audio against huge collections of texts. We could mine either for transcriptions of the audio input, i.e. texts in the same language, or translations into another language. The encoders are trained to map sentences with similar meaning close in the embedding space, independently from the language and the modality. This can include paraphrases, that is, sentences which express the same meaning but with a different wording. Therefore, we argue that adding automatically mined speech transcriptions is unlikely to improve a speech recognition system, since they require training data which are exact word-by-word transcriptions of the speech. Speech-to-text translations systems however can benefit of paraphrased output, instead of strict word-by-word translations. We therefore focus on mining speech translations.

## 4.1 Large-scale speech translation mining

We used Librivox as our set of unlabeled speech data. Librivox is a repository of open domain audio books in different languages.[3] We focused on German, French, Spanish and English audio books. Data statistics are reported in Table 4.

As English texts, we use five snapshots from Common Crawl as processed in CCNet (Wenzek et al., 2019). The texts come in paragraphs which we segment into sentences and deduplicate. This yielded about 15 billion sentences. We also mined the English audio against six languages, namely Arabic, French, Spanish, Russian, Turkish and Vietnamese. We used the same 32 Common Crawl snapshots as in Schwenk et al. (2021). The amount of sentences varies from 786 million (Arabic) to 5684 million (French). The same procedure could be also applied to all 80 languages supported by the LASER encoder.

Table 4: Librivox data statistics

|  | de | fr | es | en |
|---|---|---|---|---|
| #audio books | 633 | 257 | 343 | 13 292 |
| #hours | 3 529 | 1 535 | 1 770 | 73 511 |

**Audio segmentation** is a key element to obtain high recall in speech mining since texts in Common Crawl are usually sentences. Librivox audio books are separated into different chapters, but speech data is not segmented into sentences. Voice Activity Detection (VAD) is commonly used to segment audio, as it was done to generate LibriLight (Kahn et al., 2019) or Multilingual LibriSpeech (Pratap et al., 2020). However, those audio segments are not guaranteed to be real sentences. On one hand, it cannot be excluded that (multiple) silences appear within a sentences. And on the other hand, several sentences may follow each other without any silence in between them.

In this paper, we propose to first generate multiple plausible speech segmentations, and let the mining algorithm decide which ones are best aligned with the existing texts. Eventually, we filter the mined speech/text alignments to exclude overlapping segments. For each long audio input, theoretically containing several sentences, we run VAD with Flashlight[4] pre-trained models. This generates several detected silences in the audio. Based on these detected silences, we segment the audio into several parts under the following rules: a segment boundary is defined by two silence timestamps, a segment should be at least 3 sec long and at most 20 sec. An example of this segmentation procedure is given in Figure 3.

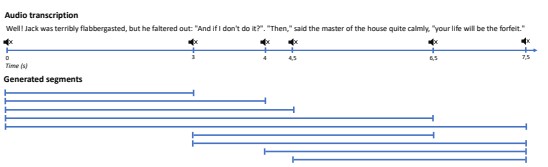

Figure 3: Example of generated segments by our segmentation. The transcription is: *Well Jack was terribly flabbergasted, | but he faltered out: | "And if I don't do it?". | "Then," | said the master of the house quite calmly, | "your life will be forfeit.".*

**Mining algorithm**. Once all audio segments and texts are encoded, we can apply mining procedures which were successfully applied to large-scale text-to-text mining, in particular CCMatrix (Schwenk et al., 2021). It has been observed that an absolute threshold on the cosine distance is globally inconsistent (Guo et al., 2018). Therefore, we apply a margin criterion as for similarity search (see Section 3.) We use the $k = 16$ nearest neighbors to calculate the average distance, for both directions. To make mining efficient at this scale, in particular when searching in fifteen billion English sentences, a compact representation and fast search is needed. The open-source FAISS library[5] for fast index search was used for this (Johnson et al., 2017), as in several other large-scale text and image mining projects.

**Post-processing**. The mining algorithm can align several sentences to the same speech segment, and vice-versa. We remove all these duplicates keeping those with highest alignment score. Our segmentation algorithm generates multiple candidates for each speech segment (see Figure 3 above). In Table 5 we report the total size of aligned speech segments (row "Sum"), and counting overlapping sub-segments only once (row "Union").

---

[3]`https://librivox.org/api/`
[4]`https://github.com/flashlight/wav2letter/`
[5]`https://github.com/facebookresearch/faiss/wiki/Faiss-indexes`

Table 5: Speech-to-text translation mined data statistics (number of hours)

| | de-en | fr-en | es-en | en-es | en-fr | en-ru | en-ar | en-tr | en-vi |
|---|---|---|---|---|---|---|---|---|---|
| Sum | 2 247 | 933 | 1 391 | 8088 | 8 327 | 3 959 | 1 718 | 1 851 | 1 527 |
| Union | 1 296 | 630 | 798 | 6825 | 7 080 | 3 529 | 1 606 | 1 721 | 1 437 |
| Post-processed | 1 074 | 543 | 668 | 6 289 | 6 544 | 3 330 | 1 549 | 1 656 | 1 390 |

We further post-process the mined data, in order to get speech segments without overlaps. We first compute a simple graph based on the following rule: two audio segments are connected if they overlap. We then compute the connected components of this graph. For each connected component, we successively select speech segments with the best similarity scores (with their mined text), provided that the speech segment is not overlapping with the previously selected segments. In particular, this post-processing ensures that similar speech segments (for example a speech segment corresponding to a full sentence, and another segment corresponding to the same sentence with an additional word at the end), are not selected twice, and that only the best matching pair is selected. It should be pointed out that we were able to align a significant percentage of the available speech, e.g. 1074 hours out of 3529 hours of raw German speech (see Tables 4 and 5).

## 4.2  Speech Translation evaluation

Finally, we train state-of-the-art `s2t` systems on the mined speech data.

**Evaluation of mined xx-En data**. We use the well established CoVoST2 task, following the train-test splits in Wang et al. (2020a). See Table 1 for corpus statistics. Initial experiments have shown that best results were obtained for a threshold of 1.07 on the alignment score. A detailed analysis is provided in the supplementary material.

The currently best performing `s2t` approach, named LNA (Li et al., 2020) builds on extensively pretrained models: a wav2vec 2.0 speech encoder (Baevski et al., 2020) and a MBART model (Liu et al., 2020) as the text decoder. MBART is first pre-trained on monolingual text data from 100 Common Crawl snapshots (Conneau et al., 2020), and then trained on the parallel texts from OPUS (Tiedemann, 2012). Li et al. (2020) jointly trained an LNA `s2t` MT system on multiple languages to enhance performance via cross-lingual transfer. Another strong multilingual `s2t` system, E2E S2T, was proposed in Wang et al. (2020a) for evaluation on the CoVoST2 dataset. It has an encoder-decoder architecture trained end-to-end. We also report the results of a cascaded model (Iranzo-Sánchez et al., 2020): the audio is first transcribed as texts, and then translated into the target language with a machine translation model.

We follow exactly the procedure of the LNA approach, but train separate models for each language pair to independently evaluate the quality of each mined speech/text corpus. We tune layer norm and multi-head attention parameters on the train set in each language direction, while other model parameters are frozen during the fine-tuning stage. The BLEU scores on the CoVoST2 test set are reported in Table 6. Our baseline bilingual LNA model is on-pair with the best performing bilingual models reported in the literature.

Table 6: BLEU scores of speech-to-text translation on CoVoST2 test set.

| Approach | Train | De-En | Es-En | Fr-En |
|---|---|---|---|---|
| **Bilingual models:** | | | | |
| Cascaded S2T | CoVoST2 | 23.2 | 31.1 | 29.1 |
| E2E S2T | CoVoST2 | 17.1 | 23.0 | 26.3 |
| LNA (ours) | CoVoST2 | 24.4 | 29.9 | 30.7 |
| LNA (ours) | CoVoST2 + mined | **26.4** | **31.6** | **32.0** |
| **Multilingual models:** | | | | |
| E2E S2T | CoVoST2 | 18.9 | 28.0 | 27.0 |
| LNA | CoVoST2 | 28.2 | 35.2 | 35.0 |

Adding the mined data brings significant improvements of more than 1.3 BLEU for all language pairs. Please note that our mined data is generic and not selected to match the domain of the CoVoST task. The results of the multilingual models are not directly comparable with ours since they benefit from knowledge transfer across languages. We provide them here for the completeness of empirical results. We aim at training a multilingual LNA s2t model with mined data in future work.

**Evaluation of Mined En-xx data.** We further evaluate the quality mined data in En-xx directions using MuST-C dataset (Di Gangi et al., 2019b) and S2T Transformer (Wang et al., 2020b), considering the established baseline results of S2T Transformer on MuST-C.

S2T Transformer used in this work has 6 encoder layers and 6 decoder layers with 4 attention heads. The feedforward dimensions are 2048 and 256. Following the empirical setup in Wang et al. (2020b), S2T Transformer is first trained on MuST-C Automatic Speech Recognition (ASR) data in order to initialize its encoder parameters. Then the model is trained with MuST-C speech translation data in a given language direction, which serves as a baseline in our experiments.

We augment the train set of speech translations with the mined data. With the encoder initialized with ASR training, S2T Transformer is trained on the combination of MuST-C and mined data for the task of speech translation for 200k steps and finetuned on MuST-C data only for 100k steps. Table 7 reports BLEU scores of models trained with and wihtout mined data in six En-xx language directions.

Table 7: BLEU scores of speech-to-text translation on the MuST-C test set.

| Train data | en-es | en-fr | en-ru | en-ar | en-tr | en-vi |
|---|---|---|---|---|---|---|
| MuST-C | 27.2 | 32.9 | 15.3 | 12.3 | 9.7 | 21.4 |
| MuST-C + Mined (t=1.07) | 28.7 | 34.4 | 16.1 | 12.8 | 10.5 | 21.8 |

As is shown in Table 7, mined data brings improvements in the BLEU score of 1.5, 1.5, 0.8, 0.5, 0.8 and 0.4, in speech translation from English to Spanish, French, Russian, Arabic, Turkish and Vietnamese respectively. The performance improvements again demonstrate that mined speech-to-text data is of good quality and useful for model training.

## 4.3 Speech-to-Speech mining

The LASER teacher text encoder and all student speech encoder are mutually compatible. This enables us to perform speech-to-speech mining directly in the embedding space without the need to transcribe or translate. We used the same speech embeddings from Librivox as in Section 4.1 and mined for all pairs of German, Spanish, French and English speech.

The amount of automatically mined speech-to-speech alignments are given in Table 8, using the same post-processing as for speech-to-text mining (applied on the source speech). Overall, we provide a speech-to-speech corpus of 1393 hours in six language pairs. This should be put into context with the current best practice in speech-to-speech translation which is mostly based on the Fisher Spanish-to-English speech corpus (Post et al., 2013) of 160 hours of source speech, the target speech being artificially created by speech synthesis (Jia et al., 2019).

We are not aware of an established framework to evaluate s2s translation systems for multiple languages. Therefore, we provide an initial human analysis of these alignments highlighting the quality of the speech-to-speech alignments. Assisted by fluent speakers in French and Spanish, we randomly sampled one hundred speech-to-speech alignments for the French-Spanish pair with alignment scores above $t = 1.06$. We manually checked the alignments and reported the accuracy we

Table 8: Speech-to-Speech mined data statistics (hours for source and target language)

| | es-de | fr-de | fr-es | en-es | de-en | en-fr |
|---|---|---|---|---|---|---|
| Sum | 64 / 65 | 52 / 59 | 235 / 259 | 732 / 936 | 557 / 821 | 1 049 / 1 210 |
| Union | 45 / 47 | 37 / 43 | 121 / 133 | 488 / 486 | 373 / 421 | 562 / 518 |
| Post-processed | 40 / 41 | 33 / 38 | 101 / 111 | 425 / 442 | 324 / 363 | 470 / 447 |

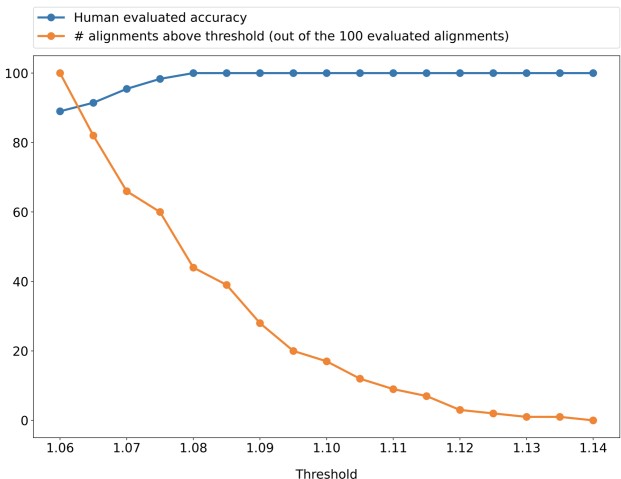

Figure 4: Human evaluation of 100, randomly sampled, `s2s` alignments for the Fr-Es pair.

obtained in Figure 4. A threshold of $t = 1.06$ corresponds to the one hundred evaluated alignments. We also report the results when removing the alignments with a score below a given threshold (varying from 1.06 to 1.14). A higher threshold gives a better accuracy but less alignments.

## 5 Conclusion

We have applied a teacher/student approach to extend an existing multilingual sentence embedding space to the speech modality. The speech encoders leverage pretrained multilingual speech representations of the XLSR/wav2vec 2.0 model. We have explored several training procedures and compared multilingual with language specific speech encoders, based on multilingual and multimodal similarity search error rate. We have empirically shown that this embedding space is suitable for large-scale speech-to-text mining. The quality of the mined speech-to-text alignments was evaluated by training state-of-the-art speech-to-text MT systems for the well-established CoVoST2 and MUST-C tasks. To the best of our knowledge, the resulting systems outperform all published results for bilingual speech-to-text MT system for translation from German, French and Spanish speech into English, and for translation of English speech into Spanish, French, Russian, Arabic, Turkish and Vietnamese.

We have also provided a proof of concept that our approach can be directly used to mine speech against speech, without the need to first transcribe or translate the audio. We hope that this method has the potential to significantly boost research in end-to-end speech-to-speech translation which currently suffers from extreme sparsity of natural training data.

One of our next steps is to extend the speech encoder to a larger set of spoken languages supported by the XLSR model. An interesting research question is whether we should use individual speech encoders for each language, or rather handle similar languages by one joint speech encoder. This similarity could be based on linguistic considerations, i.e. the same linguistic family, or rather consider phonetic similarity. In addition to extending the language coverage of the speech encoder, we can mine against large collections of texts in all the eighty languages supported by the LASER sentence encoder.

**Broader impact**. We highlight the fact that this work has potential positive impact in the society: augmenting speech datasets in an automatic and scalable way, and improving speech-related applications. At the same time, this work may have some negative consequences when the mined data is not handled in a proper way. Before using the mined data to train a speech system, one should ensure fairness in the collected data, and make sure not to train on offensive or any type of inappropriate data. We currently mine in Librivox audio books only which is not expected to contain offensive speech.

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
