# A  Appendix

## A.1   Speech Translation Evaluation

One hyperparameter in our speech translation evaluation is the threshold on the alignment scores. Mined speech-text pairs are included in the train set if their alignment scores are greater than or equal to the threshold. Speech translation models are trained on the combination of CoVoST2 train set and mined data at different thresholds. We report the performance of each model on the dev set of Common Voice in Figure 5, and find the optimal value for the threshold.

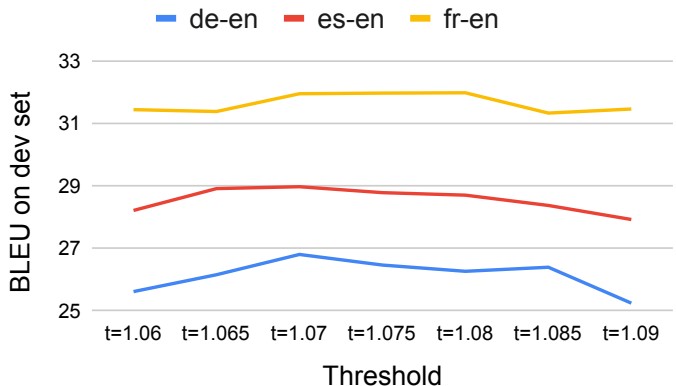

Figure 5: BLEU on dev set achieved by S2T models trained on CoVoST train set + mined data at different thresholds.

Based on Figure 5, the optimal threshold is $t = 1.07$ for de-en, es-en and fr-en directions. As we decrease the threshold, more mined data is added to the train set improving model performance. When decreasing the threshold below $t = 1.07$, the data quality decreases too: despite the larger training data size, the translation performance decreases due to more noise.

For the completeness of our analysis, we also report BLEU scores on Common Voice test set in Figure 6. Similar to our observation on the dev set, the optimal threshold is $t = 1.07$ on the test set.

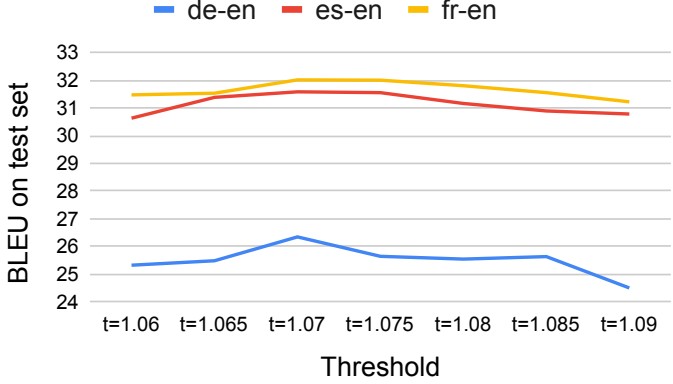

Figure 6: BLEU on test set achieved by S2T models trained on CoVoST train set + mined data at different thresholds.