# OpenReview forum: "Multimodal and Multilingual Embeddings for Large-Scale Speech Mining"
_NeurIPS.cc/2021/Conference — NeurIPS 2021 Spotlight_

### Official Review · Reviewer_1UFL · 2021-07-09

**Rating:** 7
**Confidence:** 4

**Summary:**

This paper proposes a multilingual and multimodal data mining framework that maps text and speech sentences into a shared embedding space of a fixed dimension, where sentences with similar meanings are close to each other and vice versa. Such embeddings can be then used for mining data for speech-to-speech or speech-to-text translation, which often encounter data sparsity issues.

This paper is an extension of LASER, a mining framework for only text. This paper augments LASER with a speech encoder (or multiple speech encoders, one per language) that maps an utterance into a fixed dimension representation shared with text. The authors propose to use a teacher-student framework for training the speech encoder using speech transcriptions and speech translation data. Specifically, a frozen pre-trained LASER text encoder is used, and the speech encoder to be trained is also initialized from a self-supervised speech model, XLSR. The encoder is then trained by regressing the embedding of the paired transcript or translation encoded by the LASER model. Experimental results show that 1) for some languages like Russian, using a shared encoder degrades the performance significantly, and 2) by using mined speech, performance of X-to-En translation can improve from a competitive baseline on the CoVoST2 benchmark.


**Limitations And Societal Impact:**

Yes

**Main Review:**

Strengths
- This paper is well-motivated and easy to follow. It provides a decent overview for prior work, how this work advances this line of research, high-level ideas of the proposed methods and empirical studies of how mined speech can be used.
- The proposed mining framework can be very beneficial to the speech/NLP community to tackle the data sparsity problem for speech-to-speech or speech-to-text translation. Empirical results are also presented to confirm the benefit to speech-to-text tasks.
- The authors also promise to make the mined data available to the public.

Weaknesses
- Lack of ablation studies. The authors mentioned a number of designs to improve the performance, such as 1) using a pre-trained LASER, 2) freeze the LASER encoder, 3) using a pre-trained XLSR, 4) taking embedding from the BOS frame instead of max- or mean-pooling. The readers would like to know how crucial these choices are. While running the entire mining pipeline + ST pipeline can be time-consuming, the authors could at least run the encoder evaluation (Table 2).
- While the authors argue that the trained speech encoder is compatible with the pre-trained LASER text encoder and hence can be used to mine speech for the 80 languages supported by LASER, such an example is not demonstrated empirically in the paper. This paper only shows mining in-domain language pairs (i.e., En-to-Fr/De/Es).
- The related work section is missing a number of works that learn fixed-size speech representations for a sentence. For example, CSTNet [1] learns embedding from speech translation data. A number of studies use paired video/image [2-6] and speech to learn sentence-level semantic embeddings and are evaluated with a retrieval task similar to Section 3.1.

[1] Khurana, Sameer, Antoine Laurent, and James Glass. "Cstnet: Contrastive speech translation network for self-supervised speech representation learning." arXiv preprint arXiv:2006.02814 (2020).

[2] Harwath, David, et al. "Jointly discovering visual objects and spoken words from raw sensory input." Proceedings of the European conference on computer vision (ECCV). 2018.

[3] Merkx, Danny, Stefan L. Frank, and Mirjam Ernestus. "Language learning using speech to image retrieval." arXiv preprint arXiv:1909.03795 (2019).

[4] Ilharco, Gabriel, Yuan Zhang, and Jason Baldridge. "Large-scale representation learning from visually grounded untranscribed speech." arXiv preprint arXiv:1909.08782 (2019).

[5] Harwath, David, Wei-Ning Hsu, and James Glass. "Learning hierarchical discrete linguistic units from visually-grounded speech." ICLR, 2020

[6] Monfort, Mathew, et al. "Spoken moments: Learning joint audio-visual representations from video descriptions." Proceedings of the IEEE/CVF Conference on Computer Vision and Pattern Recognition. 2021.


Questions
- For mining experiments in Section 4, is the single multilingual speech encoder used or the separate speech encoders per language?
- Why Russian data not mined for speech translation experiments? The other three languages used are relatively high-resource, and it would be interesting to see how the mined data work for low resource languages to probe how the proposed framework performs in different regimes in terms of the amount of resource.

*** EDIT

I have read the authors' rebuttal. They addressed my concerns and promised to add experiments. Hence I'm raising the score to 7

**Time Spent Reviewing:**

6

---

> ### Author Response · Authors · 2021-08-10
> **We would like to thank the reviewer for thorough review and detailed comments. Please find below our answers to the comments and questions raised.**
>
> ## Lack of ablation studies
>
> We conducted a number of ablation studies for the speech encoder, ranging from Speech Transcription teacher, Speech Translation teacher, Speech Transcription + Translation teachers to a single multilingual speech encoder and separate encoders for each language, and finally the effect of the increasing amount of training data. As mentioned in Section 3, other multilingual sentence encoders (e.g. LaBSE) could be indeed used to obtain a joint speech/text sentence embedding space. In this work, we focused on the LASER encoder since it was successfully used for large-scale bitext mining (e.g. https://arxiv.org/abs/1911.04944), is freely available and easy to use.
>
> Other architectures for neural speech processing could also be explored in order to extract a fixed-size speech sentence representation. However, we are not aware of other works on multilingual speech pre-training than XLSR.
>
> We could of course, train a new joint speech/text encoder from scratch, replacing the regression loss with a constrastive loss to ensure a good organization of the embedding space. However, we argue that is likely to require more training resources than our teacher-student approach since a new embedding space must be learned. Moreover, optimising at the same time, speech-to-speech alignments, speech-to-text alignments and text-to-text alignments from scratch remains challenging and would require its own study. Instead, our teacher-student approach benefits from the established LASER embeddings space which has shown to have interesting semantic properties.
>
> Finally, the teacher-student approach allows us to "generalize" to pairs which have not been explicitly trained. The speech encoders trained with a transcription teacher and evaluated on written translation similarity search show good performance and show that we can mine pairs which have not been explicitly trained (Table 2). As an example, we separately trained a speech encoder for French and Spanish to match the same LASER text space. This enables us to mine French/Spanish speech although this pair was never trained together. Our initial human evaluation shows the good quality of this mined pair.
>
> In the meantime, we also mined English audio against texts in eight languages (see below). These languages were not seen during the training procedure.
>
> ## Why you did not mine against all 80 languages supported by LASER?
>
> It is correct that we could mine the German, French or Spanish audio against all 80 languages supported by LASER. However, this raises the problem how to evaluate the quality of mined speech-text alignments (other than simply reporting the hours of mined data). Therefore, we chose to focus in this submission on language pairs for which well established speech-to-text (S2T) translation test sets exist and for which independent baseline results are available.
>
> In the meantime, we have also mined 70 thousand hours of English Librivox audio against eight languages (Arabic, Chinese, French, German, Russian, Spanish, Turkish and Vietnamese). This yielded a total of more than twenty thousand hours of speech/text alignment. We have started to train S2T MT systems and we will provide these results in the final version of our paper, should it be accepted.
>
> In this case, we mine against languages in the text modality which have never been seen during the teacher-student training, for instance Arabic, Chinese or Vietnamese. This is one of the advantages of teacher-student training: we only need to minimize the distance of the audio modality against one language, and we can mine against all LASER supported languages.
> Please see also our comments above on ablation studies.
>
> ## Missing related work
>
> We thank the reviewer for pointing us to these related papers. We will add a discussion in the related work section.
>
> ## Which encoder was used for mining experiments?
>
> We took the best-performing speech encoders for each spoken language on the similarity search of audio against English text. The multilingual speech encoder is used for German and Spanish, the separate speech encoder is used for French.
>
> ## Why Russian data was not mined for speech translation experiments?
>
> We had to exclude Russian audio for mining since we had not any pre-trained voice activity detection system for Russian, needed for our segmentation algorithm. We are looking into training one and will apply the same mining pipeline to Librivox Russian audio books.

---

### Official Review · Reviewer_iUuA · 2021-07-17

**Rating:** 8
**Confidence:** 4

**Summary:**

The paper presented a novel approach of encoding speech signals into representations with minimal cosine loss to multilingual text embeddings, allowing for speech data mining to generate more training data for downstream tasks. The mined data had increased the performance of state-of-the-art speech translation models by a significant margin.

**Limitations And Societal Impact:**

The authors have addressed the limitations of their approach, as well as difficulties in evaluating speech-to-speech translation tasks. Most of the societal impact of this work is positive, due to potentially increasing the performance of machine translation in low resource languages. The authors also acknowledged that mined data could contain bias or inappropriate language.

**Main Review:**

Originality: This paper introduced the new task of speech mining, which could greatly decrease the amount of human work required to generate data for the speech to text translation task; speech recognition does not benefit as much since it requires using the exact same words. The authors also mentioned the current alternative to this new task, which is to automatically generate transcriptions and discussed the improvements. The contributions and related works were stated clearly.

Quality: The authors analyzed the proposed method thoroughly, by focusing on both the speech encoder portion and how the mined data helped increase the performance in speech translation. The results were presented clearly and discussed well. Since the mined speech data should not be used as 100% accurate ground truth (for example human annotated transcriptions), the authors could perhaps have provided more information on how this influenced the quality of mined data, as well as how imperfections in the underlying multilingual embedding could introduce noise.

Clarity: The submission was well written and organized very clearly. Training details and data processing choices were discussed thoroughly as well.

Significance: The main contribution of this paper is its creativity in coming up with the new speech mining task and the benefits this could provide. The mined dataset in the paper has been shown to increase the BLEU score of the speech translation task significantly and opens up potential speech-to-speech translation as well. Although the evaluation was performed with only 3 languages, this method could be extended to all the languages the multilingual embedding supports.

**Time Spent Reviewing:**

1.5

---

> ### Author Response · Authors · 2021-08-10
> **We would like to thank the reviewer for the thorough review and detailed comments. Please find below our answers.**
>
> ## Influence of mining errors
>
> It has been argued that the LASER embeddings seem to be highly semantic, i.e. sentences which are close in the embedding space have the same **meaning**, independently of the language they are written in. This property was for instance used to find paraphrases in the same language. When mining for bitexts, it has been reported that multiple translations were found for the same source sentence, which are all valid translations.
>
> The same observations should hold for our work since we have trained the speech encoder to match the unchanged LASER text embeddings. This means that for a given speech input, other speech or text which representations are close in the embedding space, should convey the same meaning, but are not necessarily exact 1:1 transcriptions or translations.
> However, it is rather difficult to evaluate the amount and impact of this type of "noise". It could be actually beneficial to train speech-to-text (S2T) translation systems with translations which are not 1:1 translations, or with multiple correct translations for the same source speech. This is expected to make the system more robust and the translations more "natural". We are not aware of other means to evaluate the quality of mined data other than training S2T translation systems. We anticipate that for language pairs with very limited resources, slightly noisy data is better than no data, while noisy data may not be useful for resource rich language pairs.
>
> We could also use our system to search for speech with similar meaning in the same language, e.g. trying to answer the question "did someone say the same thing?".

---

### Official Review · Reviewer_YqLc · 2021-07-18

**Rating:** 7
**Confidence:** 4

**Summary:**

This paper describes an approach to project both speech and text representations into a common space.  The aim of this work is to search for new paired speech-speech instances or paired speech-text instances for translation.  This is a task that has not received much attention.  The contributions of the paper are 1) the learned representation structure and 2) the mining of data for speech to text translation demonstrating strong results.

**Ethical Concerns:**

No clear ethical issues.

**Limitations And Societal Impact:**

There aren't obvious negative societal impact to this work, outside of reinforcing biases in the training data.  This hasn't been addressed, though the positive impacts of good machine translation on text and speech have been discussed.

**Main Review:**

The strength of this work is in identifying a useful application space, and demonstrating that a common, multimodal, multilingual representation space can be used to identify useful paired data for translation.  The technical novelty is somewhat more limited, though the application technique shows substantial impact.

Specifically, the encoders (speech or text) are not novel to this work (the text encoder is pretrained). The optimization is performed via cosine similarity between the two encoders: "The margin sim(x, y) between two embeddings x and y is defined as the difference between the cosine distance between x and y, and the average cosine similarity of its nearest neighbors in both directions".

Line 94: another relevant paper on a related information retrieval task is https://ieeexplore.ieee.org/document/8059818 https://arxiv.org/abs/1701.04313.  In this work rather than enforcing a common representation for both modalities, they use a third projection network for thresholding the retrieval decision.

Consider hypothesizing or analyzing why Russian is challenging?




**Time Spent Reviewing:**

2

---

> ### Author Response · Authors · 2021-08-10
> **We would like to thank the reviewer for the thorough review and detailed comments. Please find below our answers.**
>
> ## Other related work
>
> Thank you for pointing us to the paper "End-to-End ASR-free Keyword Search from Speech" which is indeed related to our work, but considers English audio and text only and is applied to the specific task of keyword search. We will add a discussion of the similarities and differences.
>
> ## Why Russian is challenging?
>
> First of all, we have only 18 hours of training data for Russian while we have more than 100 hours for the other languages. This has of course an impact on the quality of speech encoder.
> In addition, it could be that Russian speech is more difficult to handle than English, Spanish, French or German. As an indication, the authors of the XLSR paper report a phoneme error rate of 8.1% for Russian while they obtain 2.9% for Spanish and 5.0% for French.

---

> > ### Comment · Reviewer_YqLc · 2021-09-07
> > **Thank you**
> >
> > Thank you for the responses to this and other reviews!

---

### Official Review · Reviewer_kViC · 2021-07-20

**Rating:** 7
**Confidence:** 4

**Summary:**

The authors aim to augment existing approaches of multilingual text representation, ie. embeddings, to multimodel, ie. speech/text. That is training speech/text encoders that yield the same fixed sentence-level embedding regardless of language or whether spoken or written. The paper demonstrates that the multimodal embedding approach can be used to mine a significant amount of speech in several languages from an open source LIbrivox corpus. The data that is mined was shown to improve the translation BLEU scores over a good baseline, and also yields aligned speech-to-speech data.

**Limitations And Societal Impact:**

Yes

**Main Review:**

The paper shows how multimodal/multilingual sentence embeddings can be trained on the open-source CoVoST dataset by using teacher-student training and a strong, publicly available multilingual sentence encoder called LASER. Using this multilingual/multimodal embedding system, they provide convincing results on mining data from a large audio-book corpus, Librivox, to improve speech-to-text translation system (De/Es/Fr to English).

Clarity
The paper is very readable and well organized at a high level. There are a few points where terms are mentioned without being defined/cited, but end up defined/cited later, e.g. LASER, Common Crawl, Librivox in the introduction. It would be helpful to have citations/footnotes for these when they first appear.

Quality
The authors clearly delineate the steps required to produce their results and perform speech to speech text mining. Some extra details would be helpful, like more detailed layer-by-layer descriptions of the networks involved and a per layer parameter-level accounting perhaps in the appendix. The approach taken is very sensible
- the use of teacher/student training from the LASER multilingual text encoder should generate plausible multi-model embeddings
- the modified cosine metric with nearest neighbors is a well-motivated and practical means of normalizing the score for language
- the Table 2 shows useful scores at different levels of training method.
The authors could have spent more time trying to understand why Russion multilingual speech encoder performed so poorly; if it is due to data paucity, they could have limited the amount of data in another lang and see how well the system performed. It seems more likely due to a linguistic difference (as the authors allude to in the Conclusion.

Significance
This approach is a good building block in obtaining 1 speech-to-text and 2 speech-to-speech training data from public data sources. The authors clearly show how this can be done for 1, and though they obtained data for 2, they don't demonstrate the quality of the data. The authors could have addressed this by either training s2s translation models and evaluating the results, and/or evaluating the recall rate on a known speech-to-speech corpus like Fisher En/Sp.

Minor comments:

line 68: typo this/these

line 143: cite the SpecAugment paper (DS Park).

line 148 cite BERT

Table 2, note this is similarity error rate (%). Might be useful to explicitly define an equation for the metric, it can be lost in the text of section 3.1




**Time Spent Reviewing:**

2

---

> ### Author Response · Authors · 2021-08-10
> **We would like to thank the reviewer for the thorough review and detailed comments. Please find below our answers.**
>
>  We will improve the presentation as suggested, i.e. making sure that all terms are defined as soon as they are used
>
> ## Why Russian is challenging?
>
> We have only 18 hours of training data for Russian while we have more than 100 hours for the other languages. This has of course an impact on the quality of speech encoder. We will follow the proposal of the reviewer and perform an ablation study by limiting the training resources of the other languages and compare performances. We have already conducted an initial analysis of similarity error rate in function of the speech encoder training data size, for French audio with a transcription teacher (Figure 2).
>
> It could also be that Russian speech is more difficult to handle than English, Spanish, French or German. As an indication, the authors of the XLSR paper report a phoneme error rate of 8.1\% for Russian while they obtain 2.9\% for Spanish and 5.0\% for French.
>
>
> ## Evaluation of mined speech-to-speech data
>
> At the time of submission, we had only mined data among the languages Spanish, French and German. We are not aware of speech-to-speech (s2s) translation test sets for those language pairs or baseline results.
> In the meantime, we also mined s2s data between English and these three languages, yielding for instance more than 400 hours of English-Spanish audio. We have started to train s2s translation systems and will report an evaluation and comparison to published baselines.
>
> We will also look into an evaluation based on existing speech-to-speech corpus like Fisher En/Sp.

---

> > ### Comment · Reviewer_kViC · 2021-08-30
> > **Acknowledgement**
> >
> > Thanks for taking the time to provide a response and address the comments. These additions will make the paper stronger (although I do not see a revised paper yet, the current version is dated Jun 4).

---

> > > ### Author Response · Authors · 2021-09-26
> > > **Answer to new comment**
> > >
> > > We would like to thank the reviewer for taking the time to read our comments.
> > > We have not updated the paper since we understood that the submission rules in the NeurIPS CFP do not allow to update the paper during the review process.
> > >
> > > We are providing some details below.
> > >
> > > 1) number of hours when aligning English speech against TEXTS in seven different languages:
> > >
> > >   en-es=6293h, en-de=4470h, en-fr=5864h, en-zh=1955h, en-ru=2813h, en-tr=1489h and en-ar=1320h
> > >
> > > 2) Number of hours (English) for speech-to-speech alignment involving English audio
> > >
> > > en-es=425h, de=en=324h, en-fr=470h

---

### Decision · Program_Chairs · 2021-09-27

**Decision:**

Accept (Spotlight)

**Comment:**

Reviewers unanimously voted to accepted this paper with high scores.